# Fecal Calprotectin in Patients with Crohn’s Disease: A Study Based on the History of Bowel Resection and Location of Disease

**DOI:** 10.3390/diagnostics14080854

**Published:** 2024-04-22

**Authors:** Jeongkuk Seo, Subin Song, Seung Hwan Shin, Suhyun Park, Seung Wook Hong, Sang Hyoung Park, Dong-Hoon Yang, Byong Duk Ye, Jeong-Sik Byeon, Seung-Jae Myung, Suk-Kyun Yang, Sung Wook Hwang

**Affiliations:** 1Department of Gastroenterology, University of Ulsan College of Medicine, Asan Medical Center, Seoul 05505, Republic of Korea; tonari@gmail.com (J.S.); subin1357@hanmail.net (S.S.); ssh881014@gmail.com (S.H.S.); hswooki@gmail.com (S.W.H.); umdalpin@gmail.com (S.H.P.); dhyang@amc.seoul.kr (D.-H.Y.); bdye@amc.seoul.kr (B.D.Y.); jsbyeon@amc.seoul.kr (J.-S.B.); sjmyung@gmail.com (S.-J.M.); sky@amc.seoul.kr (S.-K.Y.); 2Department of Internal Medicine, Chung-Ang University College of Medicine, Seoul 06974, Republic of Korea; 3Inflammatory Bowel Disease Center, University of Ulsan College of Medicine, Asan Medical Center, Seoul 05505, Republic of Korea

**Keywords:** Crohn’s disease, fecal calprotectin, C-reactive protein, bowel resection, disease location

## Abstract

Fecal calprotectin (FC) is commonly used to assess Crohn’s disease (CD) activity. However, standardized cut-off values accounting for bowel resection history and disease location are lacking. In this study, we analyzed data from patients with CD who underwent magnetic resonance enterography, ileocolonoscopy, and FC measurements from January 2017 to December 2018. In 267 cases from 254 patients, the FC levels in the ‘operated’ patients were higher when the disease was active compared with those who were in the remission group (178 vs. 54.7 μg/g; *p* < 0.001), and similar findings were obtained for the ‘non-operated’ patients (449.5 vs. 40.95 μg/g; *p* < 0.001). The FC levels differed significantly according to the location of inflammation, with lower levels in the small bowel compared to those in the colon. The FC cut-off levels of 70.8 μg/g and 142.0 μg/g were considered optimal for predicting active disease for operated and non-operated patients, respectively. The corresponding FC cut-off levels of 70.8 μg/g and 65.0 μg/g were observed for patients with disease only in the small bowel. In conclusion, different FC cut-off values would be applicable to patients with CD based on their bowel resection history and disease location. Tight control with a lower FC target may benefit those with a history of bowel resection or small-bowel-only disease.

## 1. Introduction

Crohn’s disease (CD) is a chronic and progressive idiopathic inflammatory disorder that affects the gastrointestinal tract. It can lead to complications such as strictures, fistulas, abscesses with loss of intestinal function, and, potentially, intestinal resection [1]. Approximately half the patients with CD require an intestinal resection within 10–20 years of diagnosis, while one-third need a second operation within the next decade [2,3]. The early identification of disease recurrence is crucial for predicting subsequent clinical course, highlighting a need for effective means to monitor disease activity in the postoperative setting [4].

A combination of ileocolonoscopy (IC), cross-sectional imaging such as magnetic resonance enterography (MRE), and clinical and biochemical markers has been adopted as the basis for CD disease monitoring [5,6]. However, repeated ICs and cross-sectional imaging are expensive, invasive, and eventually burdensome for patients with CD. Recently, fecal calprotectin (FC), a biomarker of intestinal inflammation, has demonstrated usefulness in the diagnosis and monitoring of CD, with its advantages being that it is cost-effective, non-invasive, and well tolerated compared to other methods [7,8]. Several studies have shown that FC can reflect endoscopic disease activity in CD, and it is more sensitive than the Crohn’s disease activity index (CDAI) or C-reactive protein (CRP) [9].

However, the role and precise cut-off values of FC in postoperative patients remain unclear considering the high prevalence of bowel resection among patients with CD [10]. The cut-off values suggested in various studies on patients with CD with a history of bowel resection vary widely (50–270 μg/g) [11,12]. While recent guidelines have suggested cut-off values of FC between 100 μg/g and 150 μg/g based on these studies, their limitation of incorporating only IC data, rather than including cross-sectional imaging to evaluate SB lesions, may have confounded the appropriate cut-off values for FC [13,14]. Moreover, there are conflicting reports on the diagnostic accuracy of FC regarding the influence of disease location in patients with CD [15]. Some studies have shown lower FC levels in small bowel (SB) disease compared to CD involving the large bowel (LB), while others have reported no significant difference according to location [16,17]. Considering the paucity of evidence regarding the precise cut-off values of FC in postoperative patients and those with different disease locations, we compared the performance of FC in patients with a history of bowel resection and those without while considering the disease location. 

## 2. Materials and Methods

### 2.1. Study Population

The inflammatory bowel disease (IBD) registry of the Asan Medical Center, a tertiary university hospital in Korea, has been prospectively maintained since 1997 and has previously been described in detail [18]. We retrospectively reviewed 341 cases from 328 patients with CD who had undergone MRE between January 2017 and December 2018. Patients underwent laboratory examinations and CDAI scoring during each outpatient visit. Patients with clinical and biochemical data within 30 days and IC results within 3 months of MRE were eligible for inclusion. Patients with diseases other than CD, with incomplete IC due to stenosis, or with enterostomy or colostomy were excluded. The study protocol was approved by the Institutional Review Board of the Asan Medical Center (IRB no. 2021-1515)

### 2.2. Endoscopic and Radiological Assessments

Ileocolonoscopy and MRE were performed to assess the inflammation statuses of the SBs and colons in patients with CD. For patients without previous bowel resection, SB inflammation proximal to the terminal ileum was evaluated by MRE, while terminal ileum and colonic inflammation were evaluated by both IC and MRE. The same methods were used for patients with previous bowel resection with neoterminal ileum replacing terminal ileum (Figure 1). 

The MRE scans were performed as described previously [6]. The examinations were interpreted by board-certified gastrointestinal radiologists with appropriate experience in evaluating MRE images in patients with CD. Ileocolonoscopy was performed using an endoscope (CF-HQ290L/I; Olympus Co., Tokyo, Japan). The endoscopic examinations were performed by board-certified gastroenterologists specializing in IBD. Results from IC and MRE were used to assess whether the disease was active or in remission and to determine the location of inflammation in cases of active disease. Disease remission was defined as the combination of both endoscopic and radiologic healing, without visible ulcer or inflammation associated with CD in the colon or SB on IC, and the absence of mural or peri-enteric findings of bowel inflammation on MRE, which is in line with the definition of transmural healing [19,20,21]. All images were stored in the picture archiving and communication system of the Asan Medical Center. 

### 2.3. Clinical and Biochemical Evaluations

We collected data on birth date, sex, date of CD diagnosis, smoking status, location and behavior as defined by Montreal classification, history and extent of bowel surgery, and currently administered medications. All clinical and demographic information was retrieved from the IBD registry and electronic medical records of the Asan Medical Center.

The following serum chemistry biomarkers were checked on the clinic visit day: serum albumin, normal range: >3.5 g/dL (Cobas 8000 modular analyser, Roche Diagnostics, Basel, Switzerland; AU5800 Beckman Coulter, Brea, CA, USA); serum CRP, normal range: <0.6 mg/dL (Cobas 8000 modular analyser, Roche Diagnostics, Basel, Switzerland; AU5800 Beckman Coulter, Brea, CA, USA); and ESR, normal range: 0–9 mm/h for men and 0–20 mm/h for women (TEST1, Alifax, Padova, Italy). 

Fecal calprotectin measurements were performed within 30 days of MRE date. If multiple samples were collected from one patient within the time interval, the closest result from the MRE date was used. Fecal samples collected by patients were refrigerated (2–8 °C) and sent to the department of laboratory medicine at the Asan Medical Center within one day of collection for analysis. The FC levels were measured using the Quantum Blue Calprotectin rapid test (BÜHLMANN Laboratories AG, Schönenbuch, Switzerland) per the manufacturer’s instructions. Samples with FC levels below the assay range (<30 mg/kg) were counted as 29.9 mg/kg in the analyses [19].

### 2.4. Statistical Analysis

Continuous variables are reported as medians and IQR and were analyzed using Student’s *t* test or Kruskal–Wallis test as appropriate for data normality. Categorical variables were reported as numbers and percentages and were analyzed using chi-squared test or Fisher’s exact test. Correlation analyses were performed using Pearson’s correlation coefficient. Receiver operating characteristic (ROC) curves were used to compare the diagnostic ability of FC in patients grouped according to history of bowel surgery and location of inflammation. Values of area under the ROC curve (AUROCs) and cut-off values, sensitivity, and specificity of each ROC curve were determined. The cut-off point with the best diagnostic value was determined using the Youden method for each ROC curve [22]. All statistical analyses were performed using R version 3.4.0 (R Development Core Team, Geneva, Switzerland). Statistical significance was set at *p* < 0.05.

## 3. Results

### 3.1. Patient Characteristics

From the enrolled patients, we excluded five patients with diagnoses other than CD, five with enterostomy or colostomy, four with nationalities other than South Korean, and fifty-nine without colonoscopy within 3 months of undergoing MRE. One other patient, despite satisfying the inclusion criteria, experienced severe infection at the time and was excluded. In total, 267 cases from 254 patients were included in the study. The median difference in the time of FC and IC measurement from MRE were 1 (IQR: 0–8.25) day and 2 (IQR: 2–13.0) days, respectively. No significant clinical changes were noted for any patient during that period. 

Of the 254 participants, 191 (75.2%) were male and 63 (24.8%) were female, and the median age at the time of evaluation was 27.0 (IQR: 22.0–34.0) years. The median duration of disease prior to evaluation was 5.0 (IQR, 2.0–9.0) years. The median time interval between bowel resection and the FC measurement was 3.2 (IQR: 1.1–7.8) years. Other demographic and baseline characteristics of the patients, grouped according to history of bowel resection and disease status, are summarized in Table 1. 

### 3.2. Correlation of FC with Clinical and Biochemical Markers 

The FC levels showed a weak correlation with CRP in patients with and without a history of bowel resection (r = 0.28 and *p* = 0.024 and r = 0.38 and *p* < 0.001, respectively). The FC levels correlated moderately with ESR in patients without a history of bowel resection (r = 0.45; *p* < 0.001) but not in patients with a history of bowel resection (r = 0.24; *p* = 0.059) (Appendix A). The CDAI had a weak correlation with the FC levels in patients with and without a history of bowel resection (r = 0.33 and *p* = 0.008 and r = 0.32 and *p* < 0.001, respectively) (Appendix A).

### 3.3. FC and CRP Levels in Patients according to History of Bowel Resection

The median FC levels were significantly different between the active and remission statuses regardless of bowel resection history (all, *p* < 0.001) (Figure 2A–C). Cases with previous bowel resection had median FC levels of 178 μg/g (*n* = 56) and 54.7 μg/g (*n* = 7) for the active status and remission status, respectively, while the FC levels were 449.5 μg/g (*n* = 154) and 41.0 μg/g (*n* = 50) in cases without a history of bowel resection, respectively (*p* < 0.001).

The C-reactive protein level was significantly different between the active status and remission status in all patients and in patients without previous bowel resection, although not for patients with a history of bowel resection (*p* = 0.37) (Figure 2D–F). 

There were 168 cases with positive FC levels according to conventional thresholds (≥100 μg/g) (80% of all 210 cases with active disease; 43 of 56 cases (76.8%) in operated patients; and 125 of 154 cases (81.2%) in non-operated patients) and 78 cases with positive CRP levels according to the threshold of ≥0.5 mg/dL (37.1% of all 210 cases with active disease; 20 of 56 cases (35.7%) in operated patients; and 58 of 154 cases (37.7%) in non-operated patients). 

### 3.4. Comparison of FC Levels According to Disease Location

Patients with an active disease status were further classified by disease location into subgroups labeled SB only, colon, and colon with SB (inflammation in both locations). For all 267 cases, the FC levels were significantly higher in each active disease location subgroup compared with the levels for those in remission (remission: *n* = 57, 42.6 μg/g; SB: *n* = 110, 251.5 μg/g; colon: *n* = 10, 691.5 μg/g; colon + SB: *n* = 90, 545 μg/g; *p* < 0.05 for all disease locations in comparison with remission state) (Figure 3A). Similar results were obtained for patients with a history of bowel resection (remission: *n* = 7, 54.7 μg/g; SB: *n* = 30, 148.5 μg/g; colon + SB: *n* = 26, 216 μg/g; *p* < 0.05 for both disease locations in comparison with remission state) and those without (remission: *n* = 50, 40.95 μg/g; SB: *n* = 80, 277 μg/g; colon: *n* = 10, 691.5 μg/g; colon + SB: *n* = 64, 786.5 μg/g; *p* < 0.05 for all disease locations in comparison with remission state) (Figure 3B,C).

### 3.5. Receiver Operating Characteristic Curve Analysis of FC according to History of Bowel Resection and Disease Location

For patients in any active state of disease, the AUROCs of FC for all patients, patients with a history of bowel resection, and patients without bowel resection were 0.877 (95% CI, 0.824–0.930), 0.908 (95% CI, 0.860–0.984), and 0.882 (95% CI, 0.819–0.936), respectively. The optimal FC cut-off levels to predict any active disease for all patients, patients with a history of bowel resection, and those without were 89.8 μg/g, 70.8 μg/g, and 142.0 μg/g, respectively (Table 2). 

The ROC curve analysis for the subgroup of patients with inflammation located only in the SB showed lower optimal cut-off levels of FC than for any active state of disease (66.4 μg/g for all patients, 70.8 μg/g for operated patients, and 65.0 μg/g for non-operated patients; AUROC values were 0.832 (95% CI, 0.766–0.896), 0.867 (95% CI, 0.736–0.962), and 0.837 (95% CI, 0.759–0.905) for all patients, operated patients, and non-operated patients, respectively).

An additional analysis of various cut-off values for FC, CRP, and CDAI in patients with a history of bowel resection according to the disease location was performed, and the corresponding sensitivity and specificity values were calculated (Appendix A). The AUROC values, sensitivity, and specificity for the optimal cut-off (Youden test) were much higher for FC than for CRP and CDAI. When the ROC curves for FC and CRP were compared in all patients, those with a history of bowel resection, and those without a history of bowel resection, the results of the DeLong’s test for two ROC curves showed significantly higher AUROC values for FC with *p* values < 0.001 for all groups (Figure 4).

## 4. Discussion

This study showed that FC performed better than other biomarkers of inflammation in predicting active disease in patients with CD regardless of whether they had a history of bowel resection. When classified according to disease location, the FC levels were lower in patients with SB disease compared with those with colonic or ileocolonic disease regardless of whether they had a history of bowel resection. Optimal FC cut-off levels are suggested according to the ROC curve analysis while considering both the history of bowel resection and disease location. Our results suggest that different FC cut-off values could be applied to patients with CD according to whether they have a history of bowel resection and the disease location. Additionally, tight control with a lower FC cut-off value could be considered in patients with a history of bowel resection or with disease only in the SB, conforming with the current guidelines [23,24].

Since many patients with CD undergo bowel resection at least once in their lifetimes, and recurrence is frequent in the postoperative setting, there have been efforts to validate various biomarkers to predict active disease [4,25]. An acute-phase reactant, CRP, has been known to aid in monitoring disease activity in patients with CD. However, in a postoperative setting, it is known to correlate less well with endoscopic recurrence [26,27]. A composite of eight scored subjective and objective items, the CDAI, has been used as a clinical biomarker since 1979 to support clinical evaluation and decision making [28]. Although CDAI scores <150 have been used in medical trials to indicate remission status, reports of a lack of validity in the postoperative setting suggest against using the CDAI as a primary outcome measure due to its relatively modest sensitivity [29]. 

Fecal calprotectin is known to be a sensitive surrogate marker of bowel inflammation and is used for diagnosis, to monitor treatment response, and for the early detection of disease aggravation in patients with CD [13,30]. Regarding the history of bowel resection, there have been reports of the capacity of FC to predict postoperative recurrence using IC as a reference test. Appendix A summarizes data from previous studies on FC in patients with CD with a history of bowel resection; FC levels ranging from 50 to 270 μg/g have been proposed as the cut-off values [10,11,12,31,32]. In the pivotal prospective, randomized, multicenter Postoperative Crohn’s Endoscopic Recurrence (POCER) trial, the FC levels were measured before surgery and at 6, 12, and 18 months after the resection of all macroscopic CD. The median FC levels were higher in patients with disease recurrence (Rutgeerts score (RS) ≥ i2) than in those in remission (275 vs. 72 μg/g, respectively; *p* < 0.001). Levels of FC > 100 μg/g indicated endoscopic recurrence with 89% sensitivity and 58% specificity [10]. Based on the above-mentioned studies, FC cut-off values of 100–150 μg/g in postoperative settings have been suggested in the guidelines [13,14]. Notably, previous studies on patients with CD with a history of bowel resection have used IC as the sole method for evaluating disease recurrences by focusing on the Rutgeerts score and endoscopic recurrences in anastomosis/neoterminal ileum (Appendix A), which is why their cut-off values may be inappropriate for determining mucosal healing in postoperative CD patients. As inflammation in CD can occur in the SB, which is unreachable by IC, such cases might have been overlooked in previous studies. Therefore, from our results using IC and MRE, lower cut-off values for FC compared to previous guidelines could be suggested for tight control and mucosal healing, as emphasized by the CALM study and STRIDE-II guideline [23,24]. There is no study, thus far, that has considered the role of FC with appropriate modalities regarding the concept of mucosal healing in postoperative patients with CD.

The diagnostic accuracy of FC in patients with CD with different disease locations has also been a subject of concern [11]. A study by Zittan et al. enrolled 72 patients with IBD (23 with colonic CD, 14 with isolated small intestinal CD, and 35 with UC) and assessed FC for clinical remission and endoscopic activity [33], reporting a significant correlation between FC and endoscopic activity in patients with CD, although not in cases where only the SB was involved. The optimal cut-off value of FC for discriminating between colonic disease and remission was suggested to be 100 μg/g (sensitivity 100%, specificity 67%, *p* < 0.001). However, another study regarding the same issue found no significant difference in FC levels between the different disease locations [17]. The limitation of these studies is the small number of enrolled patients, especially those diagnosed with SB CD with and without a history of bowel resection (14 and 13 patients, respectively).

Above all, the strength of this study lies in its large sample size. Compared with previous studies on FC in different disease locations, our data comprise 30 and 80 cases of SB CD among patients with and without a history of bowel resection, respectively. This large number enabled us to compare different patient groups for a more precise estimation of the FC cut-off levels. Furthermore, we utilized MRE in addition to IC for disease evaluation, allowing for the detection of lesions proximal to the neoterminal and terminal ilea. Diagnosis based only on IC may fail to include such cases, thus resulting in false-negative results. 

However, our study had some limitations. First, the study design was retrospective. Nevertheless, our IBD registry was maintained prospectively, and all clinical data were recorded immediately during each patient visit; thus, information bias was minimized. Second, most previous studies have used RS ≥ i2, while we considered the presence of disease, which corresponds to RS ≥ i1, and used this factor as the endoscopic evaluation criterion. An analysis by Liu et al. comparing the two approaches demonstrated that when RS ≥ i2 was used, the proposed cut-off value was 276 μg/g, while the cut-off values were lower at 60.2 μg/g when RS ≥ i1 was used [12]. These results are in line with our findings, and considering the mechanism of FC in detecting inflammation in the gastrointestinal tract, RS ≥ i1 might better reflect mucosal healing and its influence on FC levels in the real-world setting. Moreover, as FC is known to be correlated with an affected mucosal surface and the depth of ulceration, patients with disease confined to the SB or those with a history of bowel resection would be expected to have lower FC levels [34]. In addition, we incorporated the concept of transmural healing, which would be a more rigorous definition of disease remission. Therefore, setting lower FC cut-off values from this evidence might better aid tight control.

## 5. Conclusions

In conclusion, this study showed that different FC cut-off values could be applied to patients with CD according to whether they have a history of bowel resection and the disease location. Tight control with lower FC cut-off values could be considered in patients with a history of bowel resection or with SB-only disease. 

## Figures and Tables

**Figure 1 diagnostics-14-00854-f001:**
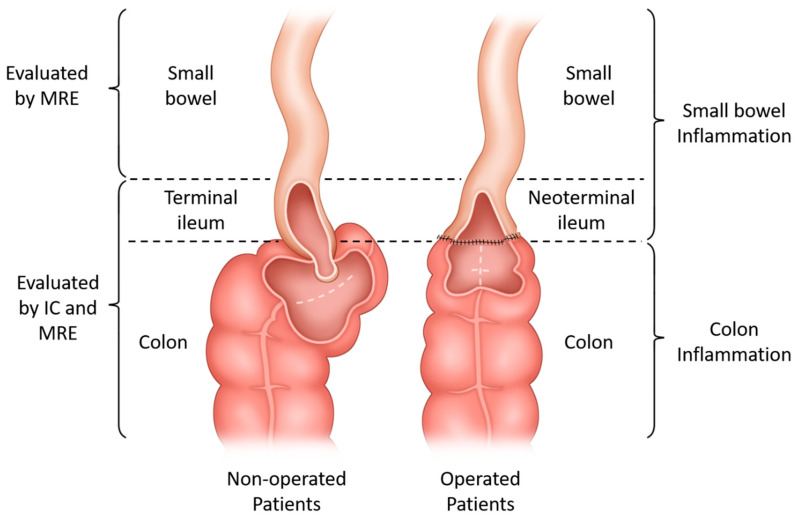
A diagram showing the study design. The two inner figures indicate the intestinal statuses among patients without a history of bowel resection involving the ileocecal valve (non-operated patients) and with a history of such surgery (operated patients), respectively. The indicators enveloping the figures show how the disease status was evaluated (left brace) and categorized for each patient (right brace). Abbreviations: MRE, magnetic resonance enterography; IC, ileocolonoscopy.

**Figure 2 diagnostics-14-00854-f002:**
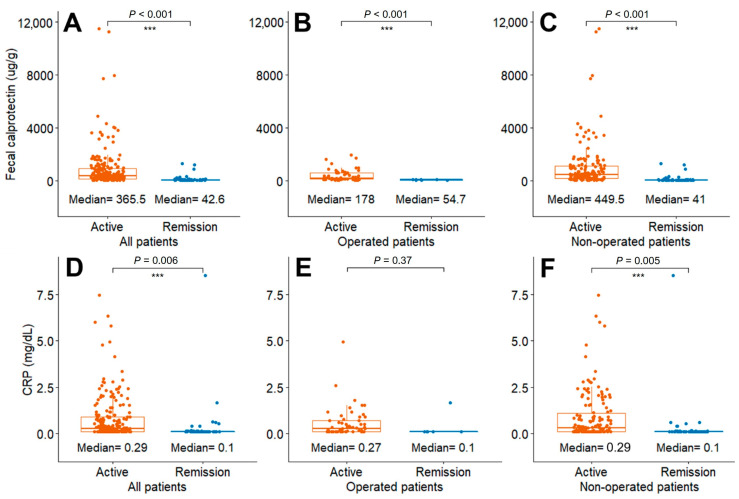
Comparison of FC and CRP levels between patients with active disease and those in remission state (**A**,**D**) all patients, (**B**,**E**) operated patients, and (**C**,**F**) non-operated patients. Abbreviations: FC, fecal calprotectin; CRP, C-reactive protein. *** *p* < 0.01.

**Figure 3 diagnostics-14-00854-f003:**
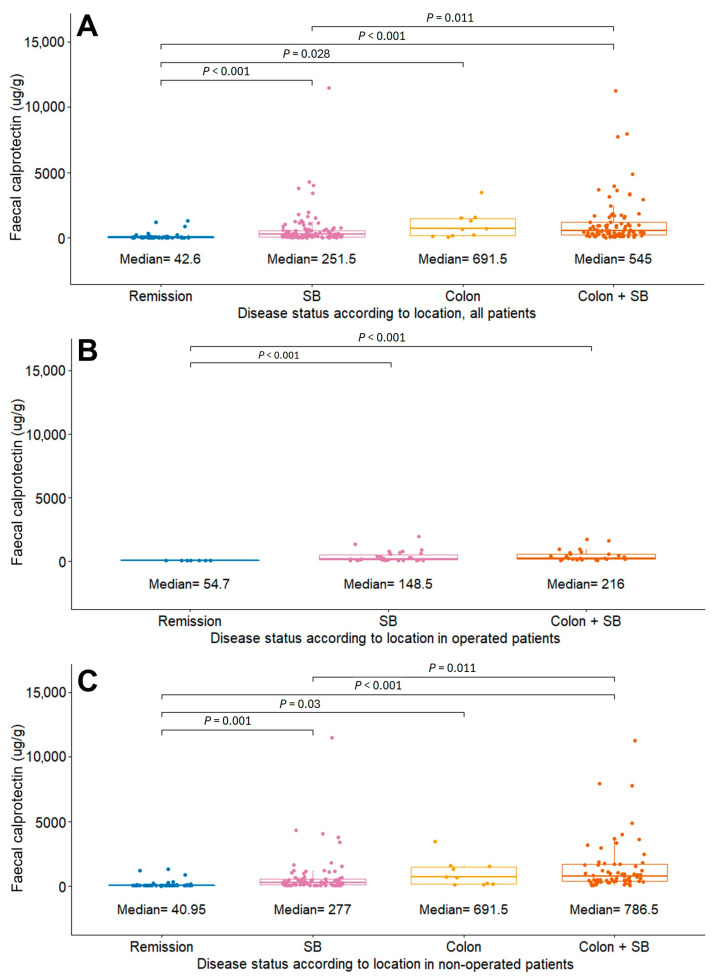
Comparison of FC levels according to disease location for (**A**) all patients, (**B**) operated patients, and (**C**) non-operated patients. Abbreviations: FC, fecal calprotectin; SB, small bowel.

**Figure 4 diagnostics-14-00854-f004:**
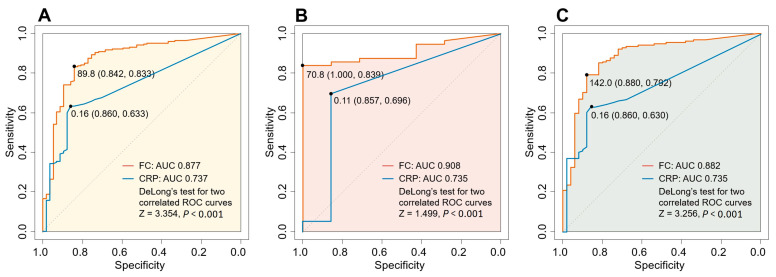
Receiver operating characteristic curve showing predictive accuracy of fecal calprotectin and C-reactive protein in discriminating between any active lesion and remission among (**A**) all patients, (**B**) patients with history of bowel resection, and (**C**) patients without history of bowel resection, respectively. Abbreviation: AUC, area under the curve.

**Table 1 diagnostics-14-00854-t001:** Clinical characteristics of study patients during evaluation (267 cases in 254 patients with CD).

	All Patients	Patients with History of Bowel Resection		Patients without History of Bowel Resection	
		Active	Remission	Active	Remission
	(*N* = 254)	(*N* = 53)	(*N* = 7)	(*N* = 144)	(*N* = 50)
Sex, *n* (%)					
Female	63 (24.8)	17 (32.1)	3 (42.9)	30 (20.8)	13 (26.0)
Male	191 (75.2)	36 (67.9)	4 (57.1)	114 (79.2)	37 (74.0)
Median age at diagnosis, *y* [IQR]	20.0 [17.0;27.0]	20.0 [16.0;29.0]	19.0 [17.0;23.0]	20.0 [17.0;27.0]	19.0 [17.0;22.0] *
Median age at evaluation, *y* [IQR]	27.0 [22.0;34.0]	29.0 [24.0;36.0]	24.0 [23.0;33.5]	27.0 [21.0;33.5]	24.0 [22.0;28.0]
Smoking status, *n* (%)					
N/A	1 (0.4)	0 (0.0)	0 (0.0)	1 (0.7)	0 (0.0)
Current smoker	57 (22.4)	12 (22.6)	0 (0.0)	38 (26.4)	7 (14.0)
Previous smoker	13 (5.1)	3 (5.7)	0 (0.0)	9 (6.2)	1 (2.0)
Never smoked	183 (72.0)	38 (71.7)	7 (100.0)	96 (66.7)	42 (84.0)
Duration of disease, *y* [IQR]	5.0 [2.0;9.0]	8.0 [3.0;13.0]	8.0 [4.5;10.5]	3.5 [1.5;8.0]	5.0 [2.0;8.0]
Duration of bowel resection, *y* [IQR]	3.2 [1.1;7.8]	3.2 [1.1;7.8]	2.8[1.8;6.9]		
Montreal behavior, *n* (%)					
B1	143 (53.6)	4 (7.5)	0 (0.0)	90 (62.5)	42 (84.0) *
B2	91 (34.1)	8 (15.1)	1 (14.3)	20 (13.9)	3 (6.0)
B3	33 (12.4)	41 (77.4)	6 (85.7)	34 (23.6)	5 (10.0)
Concurrent prescription with the below drugs, alone or in combination, *n* (%)					
5-ASA	158 (59.2)	36 (64.3)	5 (71.4)	88 (57.1)	29 (58.0)
Thiopurine	179 (67.0)	41 (73.2)	5 (71.4)	103 (66.9)	30 (60.0)
Methotrexate	22 (8.2)	4 (7.1)	0 (0.0)	15 (9.7)	3 (6.0)
Corticosteroids	7 (2.6)	0 (0.0)	1 (14.3)	5 (3.3)	1 (2.0)
Biologics	121 (45.3)	28 (50.0)	4 (57.1)	61 (39.6)	28 (56.0)
CDAI, median [IQR]	63.8 [25.9;159.5]	82.7 [43.6;193.3]	52.0 [46.2;90.1]	88.7 [28.7;176.4]	23.3 [2.0;52.5] *
FC, μg/g, median [IQR]	247.0 [70.2;685.0]	178 [108.0;563.5]	54.7 [33.7;57.1] *	449.5 [164.0;1112.0]	41.0 [29.9;78.4] *
ESR, mm/hr, median [IQR]	14.0 [5.0;27.0]	16.0 [9.5;27.0]	7.0 [4.0;14.0]	16.0 [6.0;30.0]	6.0 [2.0;11.0] *
CRP, mg/dL, median [IQR]	0.16 [0.10;0.72]	0.27 [0.10;0.71]	0.1 [0.10;0.10]	0.29 [0.10;1.15]	0.1 [0.10;0.13] *
Albumin, g/dL, median [IQR]	4.1 [3.7;4.3]	4.0 [3.6;4.1]	4.4 [ 4.3;4.5] *	4.0 [3.6;4.3]	4.3 [4.1;4.5] *
Resection type, *n* (%)					
Ileocecal resection	20 (7.9)	18 (34.0)	2 (28.6)		
Subtotal colectomy	7 (2.8)	4 (7.5)	3 (57.1)		
Right hemicolectomy	32 (12.6)	30 (56.6)	2 (28.6)		
Total colectomy	1 (0.4)	1 (1.9)	0 (0.0)		

Abbreviations: IQR, interquartile range; N/A, not available; 5-ASA, 5-aminosalicylic acid; CD, Crohn’s disease; CDAI, Crohn’s disease activity index; FC, fecal calprotectin; ESR, erythrocyte sedimentation rate; CRP, C-reactive protein; * *p* < 0.05.

**Table 2 diagnostics-14-00854-t002:** Predictive accuracy of fecal calprotectin when discriminating between active disease and remission in patients with CD.

Extent of Disease	History of Bowel Resection	AUROC	95% CI	Cut-Off Level (μg/g)	Sensitivity (%)	Specificity (%)
Any active state of disease	All patients	0.877	0.824–0.930	89.8	83.3	84.2
	Operated patients	0.908	0.860–0.984	70.8	83.9	100.0
	Non-operated patients	0.882	0.819–0.936	142.0	79.2	88.0
(Ileo)colonic disease	All patients	0.925	0.874–0.969	89.8	94.0	84.2
	Operated patients	0.956	0.868–1.000	78.4	92.3	100.0
	Non-operated patients	0.931	0.876–0.975	142.5	90.5	86.0
SB only	All patients	0.832	0.766–0.896	66.4	84.5	75.4
	Operated patients	0.867	0.736–0.962	70.8	76.7	100.0
	Non-operated patients	0.837	0.759–0.905	65.0	87.5	72.0

Abbreviations: AUROC, area under the receiver operating characteristic curve; CD, Crohn’s disease; CI, confidence interval; SB, small bowel.

## Data Availability

All data, analytical methods, and study materials relevant to this study are included in the article or are available upon request from the corresponding author, S.W.H.

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
