# Peer review of "Fecal Calprotectin in Patients with Crohn’s Disease: A Study Based on the History of Bowel Resection and Location of Disease"

_diagnostics, 2024, doi:10.3390/diagnostics14080854_

Round 1

Reviewer 1 Report

Comments and Suggestions for Authors

Fecal calprotectin (FC) is a leukocyte protein that enters the bowel lumen when inflammatory cells are shed into the lumen. It makes sense that more inflammation results in more FC being found in the gut (and ultimately in stool). The problem is that FC is measured as a "concentration" in stool (ug/g), so the results of the test are influenced by dilution by stool water. People with bowel resections tend to have larger stool weights and hence more dilution of FC and lower reported FC concentrations. This is exactly what the authors have shown. Small bowel disease generally results in higher stool weight than colonic disease because of the greater volume of water traversing the small bowel, and magnifies this dilution effect.

The problem with using cut-off values for FC is that the concentration is determined by the amount of FC entering the lumen (presumably proportional to the degree of inflammation) and stool weight (presumably greater with bowel resection). Thus there are two variables that determine FC concentration. 

In a way, it would be better to measure FC concentration and stool weight, and calculate the mass of FC excreted per day, which would be more indicative of the degree of inflammation. No one likes to collect 48-h stool samples, however, and so it is unlikely that determination of FC concentration on a spot stool sample can ever perform much better that you have reported. Your conclusion that cut-offs for FC need to be modified in the presence of bowel resection is sound, but perhaps we should change how the test is done to improve the value of measuring FC in stool.

Reviewer 2 Report

Comments and Suggestions for Authors

Overall, the research is novel and could attract wide readers in the field. I did enjoy the most reading this manuscript. It provides a new insight about proposed cut-offs values of patients with CD and related issues.

However, some minute details need to be amended to make this manuscript better.

Line no. 154 , table 1, please adjust column width for each of them as the widths are not the same and does not look fine. Also, the numbers of each column, feel free to make them in one row and align them in the same row. It appears slightly shifting up and down.

Line no. 182, figure 2, should * sign be added to any graph that expresses a significant difference? Although, readers can get this information from p-value data. From my point of view, the * (star sign) should be added there.

The rest is clear to me.

THank you for considering my opinions.
